

# Persistent La Niña's favor joint soybean harvest failures in North and South America

Raed Hamed[1], Sem Vijverberg[1], Anne F. Van Loon[1], Jeroen Aerts[1,2], Dim Coumou[1,3]

[1]Department of Water and Climate Risk, Institute for Environmental Studies (IVM), Vrije Universiteit Amsterdam, Amsterdam, the Netherlands

[2]Deltares Institute, Delft, the Netherlands

[3]Royal Netherlands Meteorological Institute (KNMI), De Bilt, the Netherlands

*Correspondence to*: Raed Hamed (raed.hamed@vu.nl)

**Abstract.** Around 80% of global soybean supply is produced in southeast South America (SESA), central Brazil (CB) and the United States (US) alone. This concentration of production in few regions makes global soybean supply sensitive to spatially compounding harvest failures. Weather variability is a key driver of soybean yield variability, with soybean especially vulnerable to hot and dry conditions during the reproductive growth stage in summer. El Niño Southern Oscillation (ENSO)

teleconnections can influence summer weather conditions across the Americas presenting potential risks for spatially compounding harvest failures. Here, we develop causal structural models to quantify the influence of ENSO on crop yields via mediating variables like local weather conditions and extratropical sea-surface temperatures (SST). We show that soybean yields are predominately driven by soil moisture conditions in summer explaining ~50%, 18% and 40% of yield variability in SESA, CB and US respectively. Summer soil moisture is strongly driven by spring soil moisture as well as remote extra-

tropical SST patterns in both hemispheres. Both of these soil moisture drivers are again influenced by ENSO. Our causal models show that persistent negative ENSO anomalies of -1.5 standard deviation (SD) lead to a -0.4 SD soybean reductions in the US and SESA. When spring soil moisture and extratropical SST precursors are pronouncedly negative (-1.5 SD), then estimated soybean losses increase to -0.9 SD for US and SESA. Thus, by influencing extratropical SSTs and spring soil moisture, persistent La Niña's can trigger substantial soybean losses in both the US and SESA, with only minor potential gains

in CB. Our findings highlight the physical pathways by which ENSO conditions can drive spatially compounding events. Such information may increase preparedness against climate related global soybean supply shocks.

## 1 Introduction

Joint soybean harvest failures in key producing regions can put substantial pressure on the global food system (Venter, 2022). The highly interconnected global food trade network means local crop failures can trigger cascading impacts effecting

commodity prices, food security and regional socio-political instability (Bren D'Amour et al., 2016; Puma, 2019; Puma et al., 2015; Torreggiani et al., 2018; Von Uexkull et al., 2016). Food crises are seldomly attributable to one single factor. Nevertheless, unfavorable weather and climate conditions can threaten the stability of the global food system. For example,



both the 2007/2008 and 2010/2011 global food crises coincided with climate-driven production shortages in major crop producing regions (Anderson, 2018; Braun, 2008; Bren D'Amour et al., 2016; Gilbert and Morgan, 2010; Timmer, 2010).

Global soybean supply is particularly sensitive to joint production failures as more than 80% of global market supply is coming from the United States, Brazil and Argentina alone (Anderson et al., 2017b; Iizumi and Sakai, 2020; Torreggiani et al., 2018; Wellesley et al., 2017). A large part of the soybean production is dedicated to animal feed (Cassidy et al., 2013). In consequence, production deficits can also affect the increasingly growing market for animal products (Cassidy et al., 2013;

Leister et al., 2015). For instance, drought conditions in 2012 in the US, Argentina and South Brazil led to important local soybean harvest failures (Elliott et al., 2018; van Garderen and Mindlin, 2022; Goulart et al., 2021; Hoerling et al., 2014). These triggered global shortages in soybean supply, affected the livestock industry and led to unprecedented increases in soybean commodity price (Leister et al., 2015; Voora et al., 2020; Zhang et al., 2018).

Soybean harvest failures are often related to anomalously hot and dry conditions in summer (Hamed et al; 2021). Studies have shown that inter-annual climate variability is one key factor that strongly impacts crop yields (Lobell et al., 2011; Lobell and Field, 2007; Ray et al., 2015). Crops are particularly vulnerable during the reproductive period, occurring in January-February-March (JFM) for South America and July-August-September (JAS) for North America (Anderson et al., 2017b, 2017a; Ortiz-Bobea et al., 2019). Hot and dry conditions can occur over large spatial domains and thereby trigger important national

production deficits (Elliott et al., 2018; Geirinhas et al., 2021; Lesk and Anderson, 2021). Such conditions are often linked to large-scale oceanic and atmospheric anomalies, which can induce spatially compounding production deficits in key soybean production regions (Anderson et al., 2017b; Heino et al., 2018; Steptoe et al., 2018).

To assess global soybean supply risks, it is essential to investigate the co-occurrence of summertime hot-dry conditions over

the main soybean producing regions. Previous work highlighted that El Niño–Southern Oscillation (ENSO) variability can drive correlated risk in North and South America soybean producing regions (Anderson et al., 2017b, 2017a, 2018, 2019). ENSO pathways that cause impacts on soybean production in North or South America can be either direct or indirect. For example, ENSO can directly affect the reproductive period soil moisture levels (i.e. in summer) and thereby crop yields (Anderson et al., 2017b). But, ENSO can also affect spring soil moisture, indirectly effecting soybean production via soil

moisture memory (Anderson et al., 2017b). Similarly, ENSO can affect extra-tropical sea surface temperature regions (SST) that can subsequently induce atmospheric teleconnections of their own which might impact crops during the summer reproductive period (Anderson et al., 2017b; MacLeod et al., 2021; Vijverberg and Coumou, 2022). Relevant examples are the north pacific SST conditions which can influence boreal summer hot and dry conditions in the US and the south Atlantic SST conditions which can influence austral summer hot and dry conditions in South America (Barros et al., 2000; Cai et al., 2020;

Doyle and Barros, 2002; Gelbrecht et al., 2021; Jorgetti et al., 2014; McKinnon et al., 2016; Sasaki et al., 2021; Vijverberg et al., 2020; Vijverberg and Coumou, 2022). Both these patterns are influenced by ENSO teleconnections which highlights a





complex chain of causes and effects that can lead to spatially compounding soybean losses (Alexander et al., 2002; Cai et al., 2020; Gelbrecht et al., 2021; Vijverberg and Coumou, 2022).

Understanding and quantifying the causal teleconnection pathways impacting global soybean production is essential to assess risks of joint crop failures under present and future climates. Nevertheless, a self-consistent framework linking the direct and indirect effects of ENSO on global soybean production is currently missing. Still, multiple studies have focused on specific sub-elements of such causal network, providing a rich scientific basis to connect relevant variables together. Here, we connect these pieces of evidence by constructing linear structural causal models that allows us to quantify direct and indirect effects

on soybean yields in the main growing regions. We first identify the direct and indirect effects of summer soil moisture, temperature and antecedent soil moisture (spring) on yield anomalies in the three regions of study separately. We then proceed to identify the main SST teleconnections affecting local summer hot and dry conditions in the three considered regions. Finally, we investigate the influence of ENSO on those SST and thereby its role in modulating spatially compounding soybean harvest failures.

**2 Material and methods**

**2.1 Soybean yield data**

Soybean crop statistics at county scale for the period 1980-2019 for the United States, Argentina and Brazil are obtained from the US Department of Agriculture (USDA), National Agricultural Statistics Service (NASS) quick stats database (https://quickstats.nass.usda.gov/), the Integrated Agricultural Information System (http://www.siia.gov.ar/) and the Brazilian

Institute of Geography and Statistics (https://www.ibge.gov.br/https://www.conab.gov.br/), respectively. A linear trend is removed from yield values at county scale to eliminate long-term effects largely due to technological improvements over the study period. Spatial information on the harvested area and the extent of the rainfed managed soybean production is obtained from the monthly irrigated and rainfed crop area database around the year 2000 (MIRCA2000), a global gridded dataset at a 0.5° resolution (Portmann et al., 2010). Counties are selected for further analysis when at least 30 out of the 40 possible county

scale yield data points are available and when rainfed agriculture constitutes at least 90% of the county scale production area share. Selected counties are grouped into three regions qualitatively defined based on existing literature on regional climatic regimes (Anderson et al., 2019; Beck et al., 2018). Soybean production counties in Argentina and south Brazil are combined into one region we refer to as the South East South American (SESA) region. The central Brazilian (CB) production counties are combined into one region and the eastern United Stated production counties into one region (US) (Fig. 1a). Wisconsin,

Michigan, Minnesota and North and South Dakota are omitted from the US cluster as these northern states were previously shown to have a different seasonal climate sensitivity compared to the rest of the US regions considered (Hamed et al., 2021; Schauberger et al., 2017). Weighted spatial average timeseries are calculated at regional level for the three defined regions





using local harvested area as weight (Fig. 1b). In this way, regionally-averaged time series emphasize major producing counties in each cluster rather than counties with less contribution to overall production.

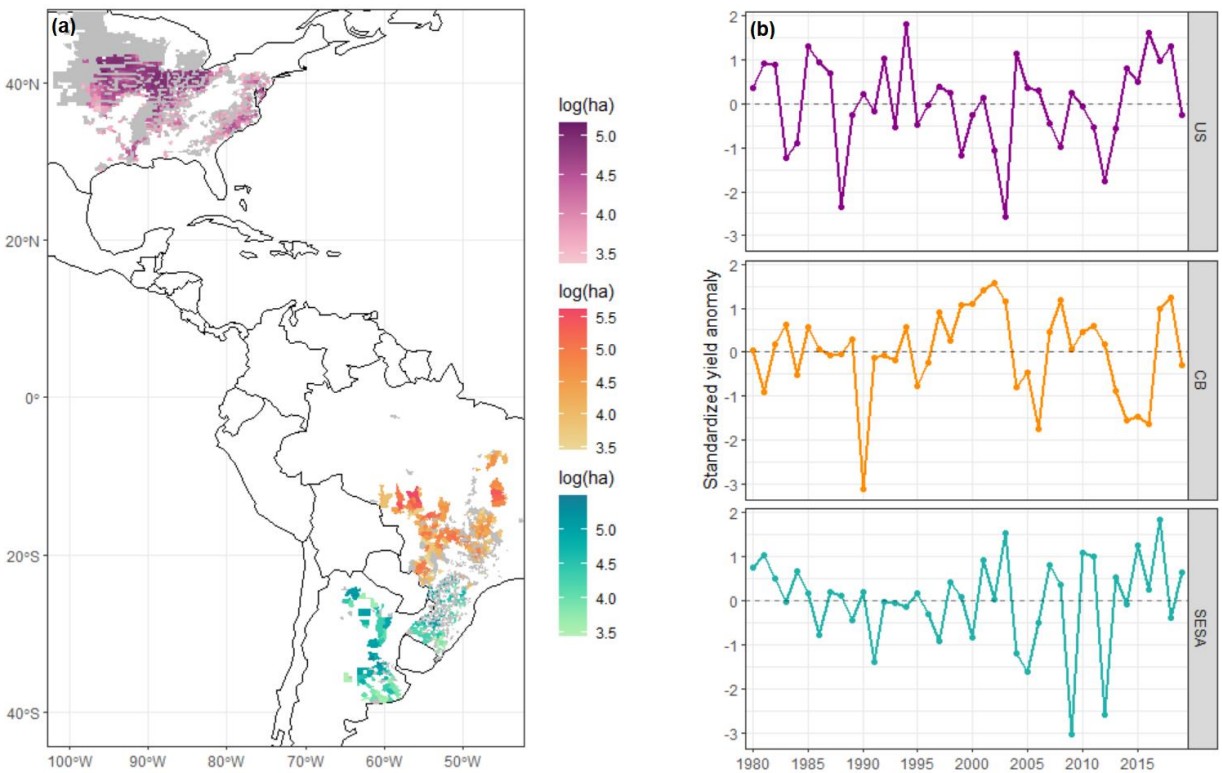

**Figure 1. Panel (a) shows soybean harvested area in hectares (ha) per county (logarithmic scale) in North and South American breadbaskets considered for this analysis. Colours represent three separate cluster regions: US (Purple), Central Brazil (Orange) and South East South America (Green). (Gray) represent masked out regions due to data availability and land management. Panel (b) shows harvest area weighted average yield time series for the three considered target regions.**

## 2.2 Climate data

The soybean main growing season extends from October to May in the SESA and CB regions and April to October in the US region (Portmann et al., 2010). In what follows, growing season conditions are split into vegetative (spring) and reproductive (summer) stages to account for the varying soybean weather sensitivities across time. The vegetative stage (spring period) is considered to extend over October, November and December (OND) for regions in the southern hemisphere and April, May and June (AMJ) for regions in the northern hemisphere. The reproductive stage (summer period) is considered to extend over January, February and March (JFM) for regions in the southern hemisphere and July, August and September (JAS) for regions in the northern hemisphere. Other studies have considered a more dynamic representation of crop developmental stages based on, for example, phenological heat units (Schauberger et al., 2017). Nevertheless, these have led to qualitatively similar results and therefore we have opted here to simply rely on fixed 3-month periods. Root zone soil moisture (SM) ($m^3/m^3$) is obtained from the modelled GLEAM v3.5a dataset that assimilates observed satellite-based soil moisture input (Martens et al., 2017).



The dataset is downloaded at 0.5 degree resolution for the period 1980-2019 and temporally averaged over aforementioned vegetative (spring) and reproductive (summer) periods. Maximum temperature (Tmax) (°C) and rainfall (mm) are obtained from the bias-adjusted WFDE5 v2.0 reanalysis dataset covering the period 1979-2019 at a daily time step and a 0.5° grid resolution (Cucchi et al., 2020). In order to isolate particularly harmful temperatures, maximum temperature is further processed into killing degree days (KDD) using 35 °C as critical temperature threshold (Tc). KDD is calculated following Eq. (1):

$$KDD = \sum_{i=1}^{n} \max(0, Tmax, i - Tc),\tag{1}$$

where $Tmax, i$ is the maximum temperature on the i-th day and n is the number of crop summer days. For simplicity, we refer to KDD as extreme heat in what follows. Rainfall is summed over the respective spring and summer periods. All three variables are initially linearly detrended at the grid cell level and at monthly time scale using monthly aggregates to remove the long-term climate change signal. In a second step, all climatic variables are spatially averaged over considered regions using a similar harvest area weighted spatial-average approach.

In order to analyze large-scale drivers of local hot and dry conditions, monthly geopotential height at 500 hPa (Z500) and sea surface temperature (SST) variables are obtained from the ERA-5 reanalysis at 1° grid resolution for the period (1979-2019) (Hersbach, 2017). These variables are detrended at the grid cell level to remove the long-term climate change signal and sequentially averaged over the respective spring and summer periods. Furthermore, 3-monthly ENSO time series are constructed based on spatial averages of sea surface temperature anomalies (SST) in the Niño3.4 region (5N-5S, 170W-120W).

**2.3 A framework for quantifying causal pathways among climate and yield variables**

We apply structural equation modelling (SEM) fitted using the robust maximum likelihood estimator to model the relative influence of spring antecedent soil moisture, summer soil moisture, rainfall and extreme heat on soybean crop yields in the three distinct regions. Then, this framework is again applied to model the influence of remote teleconnections on regional climate and soybean yields in the three regions. SEM is a form of multiple linear regression analysis that aims to quantify direct and indirect causal effects in a set of correlated variables (Fan et al., 2016). One key value of such modelling framework is that it allows to test causal assumptions implied by a user-defined directed a-cyclical graph (DAG). A DAG is a path diagram with directed arrows indicating the hypothesized structure in which information flows among a set of variables. Each path is represented by a standardized path coefficient that quantifies the marginal effect of a driving variable on a response variable. The overall model goodness of fit is assessed using the $\chi^2$ statistic with ($P\chi^2 < 0.05$) set as a cut-off significance level. The $\chi^2$ statistic tests the null hypothesis that claims no difference between observed and model-estimated covariance matrices. It follows that model goodness of fit is concluded by failure to reject the null hypothesis (Shipley, 2000). For statistical robustness, bootstrapped estimates based on 1000 draws are produced for all estimated coefficients.





## 2.4 Linking local soil moisture conditions to Z500 and SST fields

We use same-season and lagged correlation maps for Z500 and SST fields against spatially averaged summer period soil moisture for all regions separately. Due to multiple significance testing, we account for the false discovery rate using the Benjamini/Hochberg correction and consider a corrected significance threshold set at ($P_{FDR}$ <0.05) (Benjamini and Hochberg, 1995). The strongly correlating regions that are found are in line with the current literature on links between large scale teleconnections and local summer climate variability in the US, SESA and CB. These regions are encapsulated by a bounding box and used to calculate spatial covariance timeseries between the SST anomalies and the SST correlation pattern, rendering a 1-d timeseries. The correlation pattern is weighted by the inverse of the p-value ($P_{FDR}$) to emphasize strongly correlating grid cells (Vijverberg and Coumou, 2022). These time-series are meant to present a summary of the highlighted SST pattern and are used to model the direct relationship between large-scale SST variability and local soil moisture variability.

## 3 Results

### 3.1 Impacts of hot and dry conditions on soybean yields

Using SEM, we test whether the data can reject or confirm hypothesized causal pathways. A central causal hypothesis, common to all three distinct production regions, assumes that yield variability is driven primarily by extreme heat and dry soils during summer (JAS for the US, JFM for the SESA and CB regions) (Fig. 2). We further assume that soil moisture during summer is influenced by same season rainfall variability and antecedent (spring) soil moisture conditions (AMJ for the US, OND for the SESA and CB regions). Using SEM, we show that those causal assumptions are consistent with the data in the three different regions ($P$-value for $\chi^2$ test = 0.51, 0.2, 0.12) for CB, SESA and US respectively. Specifically, the effects of spring soil moisture and summer period precipitation on crop yields are, to first order approximation, fully mediated by summer soil moisture in all regions, and thus only indirectly effect crops.





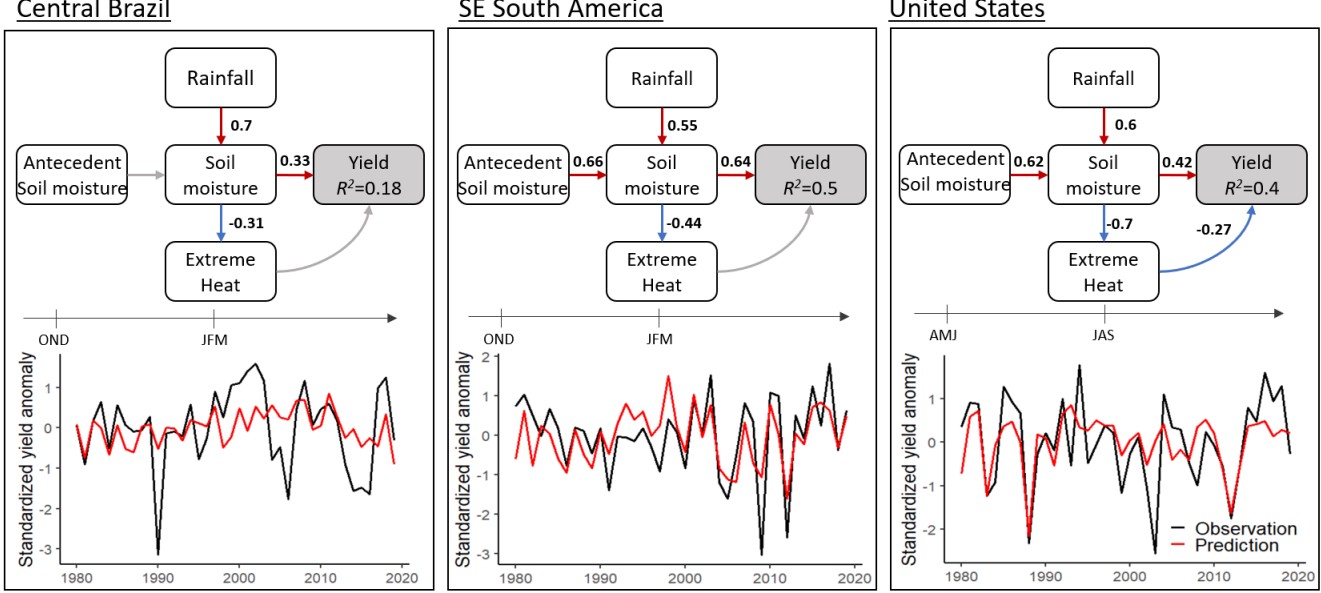

**Figure 2. Structural equation models quantifying the impacts of hot and dry conditions on crop yields in Central Brazil (left), SE South America (centre) and United States (right). Red arrows show positive while blue arrows show negative significant relationships (P < 0.05) between variables. Gray arrows show statistically insignificant relationships (P > 0.05). Observed yield timeseries are displayed in black and predicted yield time-series in red.**

Extreme heat and soil moisture during summer explain 18%, 50% and 40% of soybean yield interannual variability in the CB, SESA and US regions respectively. Summer soil moisture is found to be the main driver of yield variability in all three regions (path coefficients of 0.33, 0.64 and 0.42, all statistically significant $P < 0.05$) for CB, SESA and US regions, respectively. Extreme heat has a considerably weaker effect (path coefficients -0.18, -0.13 and -0.27) for CB, SESA and US regions, respectively, with only the US link statistically significant ($P < 0.05$). Extreme heat itself is strongly and significantly ($P < 0.05$) driven by same season soil moisture in all regions (path coefficients of -0.31, -0.44 and -0.7) for CB, SESA and US regions, respectively, suggesting the importance of land-atmosphere coupling. Spring soil moisture and summer precipitation equally influenced summer soil moisture in the US and SESA regions (path coefficients of 0.6 and 0.55 respectively, both statistically significant $P < 0.05$). In CB, spring soil moisture is not a significant driver of summer soil moisture, with only a significant contribution from summer precipitation to same season soil moisture conditions (path coefficient of 0.7, $P < 0.05$). Bootstrapped estimates for aforementioned path coefficients are presented in Fig. A1a.

## 3.2 Large-scale ocean and atmospheric drivers of hot and dry conditions in South America

We consider large-scale climate drivers of summer soil moisture given its prominent role in affecting yield variability in all considered regions (Fig. 2). To identify relevant large-scale climate drivers, we calculate same season and lagged correlation maps for both SST and Z500 anomalies with summer period soil moisture (Figs. 3a-d, 4a-b). Significant correlations are found between SST areas and summer period soil moisture for all considered regions. For Z500, significant correlations ($P_{FDR} < 0.1$)



are only found for the SESA region (Fig. A2). Nevertheless, both SST and Z500 correlation maps show physically plausible and consistent patterns and therefore we interpret them jointly, using relevant teleconnection literature.

Figure 3. Panels (a, b, c & d) show correlation maps at both lagged and same season timesteps for Z500 (contours) and SST (colours) against austral summer (JFM) soil moisture. Contour lines represent Z500 correlations with 0.1 increment where negative values are dashed. Panels (e, f & g) show correlation maps for lagged OND ENSO 3.4 index against austral summer SST (JFM), soil moisture (JFM) and yield anomaly in CB and SESA. Stippling indicates statistical significance ($P_{FDR} < 0.05$) after having corrected for false discovery rate using the Benjamini/Hochberg correction. The green rectangles in b and d indicate the SST areas used to calculate the spatial covariance timeseries.

The correlation maps for the SESA and CB regions show, to first order, opposite patterns for both SST and Z500 (Fig. 3), in particular for the same-season correlation maps (Fig. 3b-d). These opposite patterns between the two regions are in line with the well documented south Atlantic convergence zone (SACZ) activity and characterize the dominant dipole pattern of summer moisture variability over subtropical south America (Boers et al., 2014; Gonzalez and Vera, 2014; Rodrigues et al., 2019). A





suppressed SACZ event is associated with a high pressure system over eastern South America which induces low soil moisture levels over the CB region (Rodrigues et al., 2019). Concurrently, such pattern reinforces the southward direction of the South American low level jet (SALLJ), which is the main channel of moisture transport from the Amazon to the subtropics, leading to higher moisture levels in the SESA region. During an enhanced SACZ event, the opposite happens, with a low pressure

system over eastern South America channeling moisture from the Amazon eastwards towards the CB region, in turn, reducing moisture transport towards the SESA region (Boers et al., 2014; Gelbrecht et al., 2021; Gonzalez and Vera, 2014; Montini, 2019; Rodrigues et al., 2019). The southern and eastward directions of moisture transport from the Amazon into the subtropics characterize the two main SALLJ regimes (Boers et al., 2014).

The SACZ position and intensity are linked to the south Atlantic SST conditions, although some studies also highlight that

SACZ variability can itself force south Atlantic SST conditions (De Almeida et al., 2007; Chaves and Nobre, 2004; Coelho et al., 2016; Doyle and Barros, 2002; Jorgetti et al., 2014; Seager et al., 2010; Zilli et al., 2019). Here, we find a summer dipole pattern of strong and significant SST correlations in the south Atlantic with opposite effects on SESA and CB soil moisture (Fig. 3b, 3d). This SST pattern shows high correspondence with the atmospheric circulation over the eastern part of south

America and both patterns are reminiscent of previously discussed SACZ activity. We find that a high (low) pressure system over eastern south America coupled with adjacent warm (cold) south Atlantic SST conditions favor high (low) soil moisture over SESA and low (high) soil moisture over the CB region (Fig. 3b-d). Hints of such a pattern can be detected in the lagged correlation maps (Fig. 3a-c), but the relationship is clearly stronger for same season correlation maps in line with the peak SACZ activity in austral summer (Jorgetti et al., 2014). We consider the austral summer SST correlation pattern within the

green rectangles (Fig. 3b-d) as representative of the SACZ activity and associated low level jet regimes, incorporating both the SACZ atmospheric imprint on SST and the potential effects of south Atlantic SST on the SACZ. We summarize the south Atlantic SST pattern into a 1-d timeseries by calculating a spatial covariance timeseries for both SESA and CB regions separately. The 1-d summary timeseries for both regions are perfectly anti-correlated (Fig. S1). This reinforces the adequacy of the highlighted south Atlantic SST dipole pattern, hereafter South Atlantic pattern (SA), to represent the dominant opposite

signal in moisture variability over CB and SESA regions.

In addition to the SA pattern, we find contrasting correlations between CB and SESA in both the tropical Atlantic and tropical Pacific. Our results highlight that warmer SSTs in the tropical Atlantic, in particular during the austral spring period, are associated with increased moisture in the CB and decreased moisture in SESA (Fig. 3a-c). SST in the tropical Atlantic affects the position of the intertropical convergence zone (ITCZ) where cold (warm) SST strengthens (weakens) the southward shift

of the ITCZ (Cai et al., 2020). A southwards located ITCZ is typically associated with a southward SALLJ flow towards the subtropics leading to increased moisture in the SESA and decreased in CB (Gelbrecht et al., 2021). ENSO anomalies on the other hand trigger a pattern of stationary Rossby waves, referred to as the Pacific–South American (PSA) pattern that we also find in the Z500 correlation lines extending from the pacific to the Atlantic basin (Fig. 3a-d). These are most pronounced for





the SESA region and seem to originate from the tropical pacific in austral spring (Fig. 3c) compared to a more southern origin
close to Indonesia in austral summer (Fig. 3d). These south pacific wave trains in spring can impact the low-level jet, SST
conditions in the south Atlantic and the SACZ position, in turn playing a key role in modulating summer moisture variability
in south America (Gelbrecht et al., 2018, 2021; M Silva et al., 2009).

We explore the effects of ENSO on austral summer climate variability and crop yields in South America by calculating grid-
based correlation maps of the ENSO3.4 index versus summer SST (Fig. 3e), summer soil moisture (Fig. 3f), and soybean yield
(Fig. 3g). Two ENSO time-series, averaged over OND and JFM, are considered to capture lagged and same season ENSO
effects on summer conditions. Both lagged (Fig. 3e) and same-season (Fig. S2) correlation maps show similar ENSO effect
on all variables considered. The similarity between the correlation patterns of ENSO-SST (Fig. 3e) and Soil moisture-SST
(Fig. 3b, 3d) suggests that ENSO plays a significant role in influencing the austral summer SA pattern which has direct effects
on local summer soil moisture conditions. Lagged and same-season ENSO correlations with summer period soil moisture
(JFM) and yield anomaly report positive correlations between 40°S and 25°S and negative correlations between 20°S and 5°S
(Fig. 3f, 3g). This is consistent with the initial consideration of two distinct production regions in south America, in addition
to the opposite effects of the SA pattern on CB and SESA regions highlighted above.

ENSO time-series for both seasons show a stronger relationship with soil moisture compared to yield anomaly. Assuming that
summer soil moisture is the direct causal driver of local yield anomalies, weaker ENSO correlations against yield anomaly can
be understood as loss of predictive signal along the causal chain. To further explore this, we calculate grid-based correlation
maps for crop yields, the SA pattern and antecedent (spring) soil moisture against summer period soil moisture (Fig. A3). As
expected, we find that summer soil moisture is the strongest predictor of soybean yields at grid-cell level. Furthermore, we
find that the SA pattern is a stronger predictor of summer period SM compared to the ENSO index. We interpret the stronger
correlations between pairs of variables (summer SM-crop yields) and (SA pattern-summer SM) as preliminary evidence for
the, primarily, indirect effect between ENSO and summer SM. This suggests that ENSO effects on soybean yields, are to a
large extent, mediated by the relationships between (ENSO-SA pattern) and (ENSO-antecedent SM).

### 3.3 Large-scale ocean and atmospheric drivers of hot and dry conditions in the US

SST and Z500 correlation maps for summer period soil moisture in the US show a resembling pattern at lagged and same
season time steps (Fig. 4a-b).





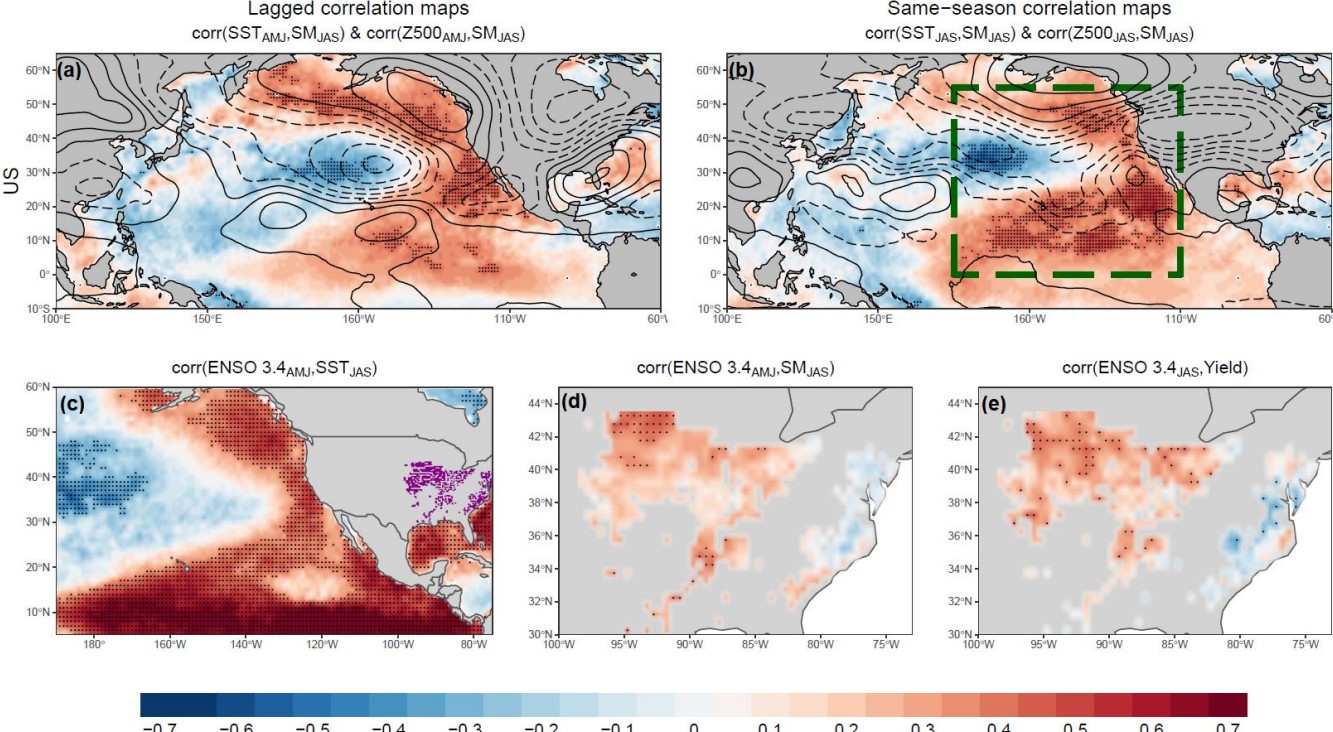

**Figure 4.** Panels (a & b) show correlation maps at both lagged and same season timesteps for Z500 (contours) and SST (colours) against boreal summer (JAS) soil moisture. Contour lines represent Z500 correlations with 0.1 increment where negative values are dashed. Panels (c, d & e) show correlation maps for lagged AMJ ENSO 3.4 index against boreal summer SST (JAS), soil moisture (JAS) and yield anomaly in the US. Stippling indicates statistical significance ($P_{FDR} < 0.05$) after having corrected for false discovery rate using the Benjamini/Hochberg correction. The green rectangle in (b) indicates the SST area used to calculate the spatial covariance timeseries.

We find a significant SST dipole pattern in the north Pacific which is similar to the Pacific Decadal Oscillation (PDO) pattern. In particular, our results show that low soil moisture in the US region is associated with a PDO-like negative state that persists throughout the growing season (Fig. 4a-b). The Z500 contour lines show an arching Rossby wave structure that originates from the pacific towards the North American continent in both boreal spring and summer (Fig. 4a-b). The boreal spring arching wave has a center of origin close to the tropical pacific region and shows strong resemblance to the typical-ENSO forced teleconnection. The summer Rossby wave has a more zonal structure and does not seem connected to tropical regions. Previous research showed that ENSO forces an extra-tropical PDO-like SST pattern in winter and spring, which in turn drives Rossby waves in summer and that carries significant predictive skill for summer hot and dry conditions in eastern US (McKinnon et al., 2016; Vijverberg et al., 2020; Vijverberg and Coumou, 2022).

We summarize the north pacific SST dipole pattern into a 1-d timeseries, hereafter the north pacific pattern (NP), by calculating a weighted spatial covariance timeseries based on the highlighted SST area in green (Fig. 4b). The 1-d timeseries highly correlates with the PDO index (correlation of 0.73; Fig. S3) confirming the visual resemblance to the PDO pattern.





Additionally, the SST correlation maps highlight that cold SST conditions in the tropical Pacific favour low summer soil moisture in the US. This is particularly the case in spring where significant correlations are present in the tropical pacific (Fig.
4a). The relevance of the tropical SST conditions in spring along the ENSO-like teleconnection highlighted in the Z500 correlation map suggest a significant ENSO role in affecting summer soil moisture in the US.

In a parallel assessment to the South American regions, we calculate grid-based correlations for summer SST (JAS), summer soil moisture (JAS), and soybean yield against the ENSO 3.4 index in AMJ (Fig. 4c-e). Correlation maps based on the ENSO
3.4 index in JAS give largely similar effects (Fig. S4). ENSO timeseries in both AMJ and JAS are significantly correlated to a summer SST pattern in the north Pacific that is very similar to the NP pattern (Fig. 4b-c). Over most regions, the ENSO index is positively correlated with summer soil moisture and yield anomalies. Only in the most eastern part, summer soil moisture is negatively correlated with spring ENSO (AMJ) (Fig. 4d-e). This is in agreement with the spatial pattern of the Z500 AMJ correlation lines against summer period soil moisture which shows opposite pressure systems following a wave
train pattern (Fig. 4a).

ENSO can equally well predict summer soil moisture (Fig. 4d) and yield anomaly in the US (Fig. 4e). The relatively high correlation between (ENSO-crop yield) compared to the correlation between (ENSO-summer SM) can be related to the unaccounted causal chain linking ENSO to extremes temperatures and finally yield anomalies. Extreme temperatures have a
significant impact on crop yields in the US region but not in the CB and SESA regions (Fig. 2). We calculate grid-based correlation maps for crop yields, the NP pattern and antecedent (spring) soil moisture against summer soil moisture (Fig. A4). Similar as in the South American analyses, we find that summer soil moisture is the main driver of soybean yields at the grid-cell level and that the NP pattern is a stronger predictor of summer soil moisture than the ENSO index. In line with previous work (Vijverberg and Coumou, 2022), we hypothesize that ENSO indirectly affects summer soil moisture by modulating the
NP pattern in from winter to summer. In the next section, we explicitly test for direct and indirect ENSO influence on summer soil moisture in all considered regions.

### 3.4 Causal model linking ENSO to yield via extratropical SST and soil moisture.

The analyses above show that ENSO has a central role in modulating soil-moisture conditions and thereby yields in the three soybean regions. Our preferred physical interpretation is that ENSO impacts spring soil moisture directly, and summer soil
moisture conditions only indirectly via extra-tropical SSTs. We now apply SEM to test whether the data can confirm this physical hypothesis (Fig. 5).





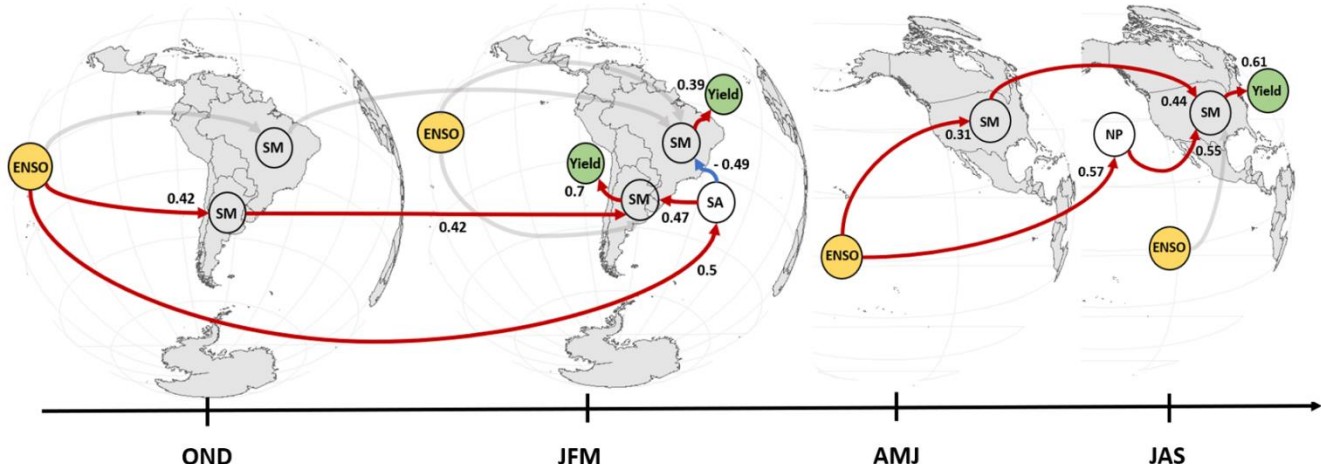

**Figure 5. Structural equation models showing assumed hypothesis linking ENSO variability in OND, JFM, AMJ and JAS to yield anomaly in CB, SESA and US regions. Red arrows show positive while blue arrows show negative significant relationships (P < 0.05) between variables. Gray arrows show insignificant relationships (P > 0.05). SM stands for soil moisture, SA for south Atlantic pattern and NP for north Pacific pattern.**

Causal assumptions implied by the network are consistent with the data in all regions ($P$-value for $\chi^2$ test = 0.07, 0.37, 0.18) for CB, SESA and US respectively. To first order approximation, this implies that the effects of large-scale SST drivers and antecedent soil moisture conditions on local crop yields are mediated by summer soil moisture. With respect to ENSO, we find that the effects of austral/boreal spring ENSO on summer soil moisture are fully mediated by antecedent soil moisture and the extratropical SST pattern. Furthermore, we do not find a statistically significant link between austral/boreal summer ENSO and summer soil moisture. Thus, we do not find evidence for a direct link from ENSO on summer soil moisture. ENSO during austral/boreal summer also does not significantly effect same-season extratropical SA/NP patterns which suggests that summer extratropical SSTs are primarily driven by spring ENSO conditions. We find that austral/boreal spring ENSO has a significant positive effect on same-season spring soil moisture conditions for the SESA (path coefficient of 0.42) and US (path coefficient of 0.42) regions but not for the CB region. Additionally, we find that austral spring ENSO (OND) is a strong predictor of the SA pattern in JFM (path coefficient of 0.5). Likewise, boreal spring ENSO (AMJ) is a strong predictor of the NP pattern in JAS (path coefficient of 0.57). The SA pattern predicts an opposite signal of soil moisture over CB and SESA with path coefficients of -0.49 and 0.47, whereas the NP pattern is a strong predictor of US summer soil moisture (path coefficient 0.55). Bootstrapped estimates for aforementioned path coefficients are presented in Fig. A1b.

We find that ENSO plays a significant role for SESA and US. It does so via affecting spring soil moisture conditions and regional extra-tropical SST patterns (the SA and NP patterns) in summer, which subsequently affect summer soil moisture. From these results, it follows that persistent La Niña states enhance the risk of spatially compound soy yield failures. Using our causal graphs, we estimate that a persistent ENSO index of -1.5 SD in austral and boreal spring leads to a -0.41, -0.43 and +0.14 SD change in soybean yields in US, SESA and CB respectively. The estimated impacts on soy yields are larger for a -



1.5 SD change in both antecedent soil moisture and extratropical SST patterns with a -0.9, -0.93 and +0.28 SD change predicted in soybean yield. These larger predicted impacts based on antecedent soil moisture and the extratropical SST are expected given the direction in which information flows in our proposed causal diagram.

## 4 Discussion

We use structural equation modelling to quantify the relationship between soybean yields and extreme heat, rainfall, and soil moisture in the US, SESA and CB regions. We take a physics-guided approach to construct a causal diagram that visualizes and quantifies the hypothesized causal structure in the data. We assume summer extreme heat and soil moisture to be the direct drivers of soybean yields in the three regions. Furthermore, we assume that summer soil moisture is directly influenced by antecedent (spring) soil moisture and same-season rainfall. Finally, we assume that extreme heat in summer is effected by same-season soil moisture conditions via land-atmosphere feedbacks. Model goodness-of-fit tested via the chi-square statistic suggests the plausibility of the hypothesized causal structure. We note that previous works have shown that soybean yields can also be impacted by excessive rainfall and soil moisture (Li et al., 2019). Future studies can explore the direct negative effects of high rainfall on crop yields in conjugation with the indirect positive effects via replenishment of soil moisture.

We find that summer soil moisture captures most of the hot and dry climate signal in soybean yield variability for all regions. This does not mean that extreme heat on its own is not an important crop stressor. In fact, leaf-scale experiments show that drought and heat limit crop growth via distinct physical pathways (Rizhsky et al., 2002, 2004; Suzuki et al., 2014). Nevertheless, here we show that extreme heat itself is strongly driven by moisture deficits in summer. Moisture and temperature variables are linked by atmospheric circulation and land-atmosphere couplings in the climate system where both can co-occur and induce reinforcing physical feedbacks (Lesk et al., 2021; Seneviratne et al., 2010). The strong atmospheric coupling is typically characterized by a persistent high-pressure system creating clear skies that result in high temperature and dry conditions and in turn lead to fast depletion of soil moisture (Basara et al., 2019). Land-atmosphere feedbacks can drive temperatures up due to a lack of cooling by evapotranspiration when soils are dry, which can further exacerbate hot and dry conditions (Lesk et al., 2021; Seneviratne et al., 2010; Sippel et al., 2016, 2018).

Here, we interpret the summer period soil moisture variable as indicative of such compound hot and dry conditions with both land-atmosphere feedbacks and atmospheric circulation mechanisms potentially at play. We find that summer soil moisture is strongly preconditioned by antecedent soil moisture in the SESA and US regions. Soil moisture memory can persist over several months which implies that spring anomalies can influence summer soil moisture conditions and in turn intensify summer land-atmosphere feedbacks (Anderson et al., 2017a; Hamed et al., 2021; Sippel et al., 2018; Sippela et al., 2016). The lack of a significant relationship between antecedent (spring) and summer soil moisture in the CB region could be related to local land-atmosphere feedbacks, which can force an inverse relationship between austral spring and summer moisture





conditions particularly during ENSO events (Grimm et al., 2007). Low moisture and high temperature conditions during spring

over central-east Brazil can increase moisture flux from the Amazon into the CB region leading to higher moisture conditions in summer (Cai *et al* 2020). We note that the soybean yield model for the CB region has significantly less predictability compared to the other two regions, which highlights that further investigation is needed in this region for more robust results.

We show that a dipole pattern in the south Atlantic drives an opposite signal in austral summer soil moisture (JFM) in the CB

and SESA regions. This pattern is closely related to the SACZ position and intensity in addition to the SALLJ regime, both of which are key mechanisms that explain the opposite moisture anomalies in the CB and the SESA regions during austral summer (Boers et al., 2014; Gelbrecht et al., 2021). This dipole pattern has been shown to be caused by southern hemisphere Rossby waves which are dominated by the Madden-Julian Oscillation variability at an intra-seasonal timescale and ENSO variability at an interannual timescale (Barros et al., 2008; Cai et al., 2020; Cunningham and Cavalcanti, 2006; Drumond and Ambrizzi,

2006; Gelbrecht et al., 2021; Rodrigues et al., 2019).  We show that a dipole pattern in the north pacific drives boreal summer soil moisture variability (JAS) in the US. This is in line with previous research that showed that the north pacific SST pattern causes a Rossby wave structure that forces a persistent high-pressure system over large parts of the US (McKinnon et al., 2016; Vijverberg and Coumou, 2022). The NP pattern can be detected already in boreal spring and arguably even earlier as ENSO teleconnections in boreal winter and spring reinforce the NP pattern via the so-called atmospheric bridge (Alexander et al.,

375 2002).

In line with previous research, we consider the ENSO phenomenon as an overarching causal driver which influences soybean growing seasons in both North and South America (Anderson et al., 2017b, 2017a, 2018). Here we show that ENSO conditions are particularly impactful in spring in South and North America, acting directly on antecedent (spring) soil moisture and the

SA and NP patterns. The value of considering ENSO effects on summer soil moisture, and in turn crop yields via mediating variables such as the extratropical SST conditions and antecedent soil moisture, allows for a more detailed description of conditions that can lead to spatially compound crop failures in North and South America. As the extratropical SST conditions can vary independently of ENSO, accounting for them explicitly gives potentially higher predictability for associated simultaneous production failures. Similarly, it can explain why not all ENSO events translate into impacts over the respective

soybean regions.

We find non-significant ENSO effects in austral and boreal summer over South and North America, respectively. Previous research shows that southern hemisphere ENSO teleconnections are very weak in summer due to the weak equator to pole temperature gradient and dominant role of the SACZ over the summer circulation in south America (Cai et al., 2020; Cazes-

Boezio et al., 2003). Similarly, weak ENSO conditions during boreal summer in addition to the dominant influence of the north pacific on summer moisture variability in the US can explain the weaker ENSO teleconnection in summer (Vijverberg and Coumou, 2022). Here, we consider El Niño and La Niña events to have symmetrical impacts which emphasizes dynamics



related to a typical ENSO event. Nevertheless, we note that diversity in impacts between phases has been previously shown (Cai et al., 2020). Moreover, different ENSO flavors have been highlighted such as the east pacific or central pacific regimes,

which can lead to different global teleconnections (Strnad et al., 2022). Furthermore, other modes of summer moisture variability can modulate the impacts of ENSO on considered soybean producing regions. For instance, the south Atlantic dipole mode, Indian ocean variability, and the south annular mode can all affect climate variability in south America during the soybean growing season (Cai et al., 2020) . Nevertheless, here we focus on ENSO given its global impacts on both north and south hemispheres in addition to its documented causal influence on aforementioned modes of climate variability (Cai et al.,

2020; Ham et al., 2021; Sun et al., 2017).

Finally, we show that La Niña conditions favor significantly lower yields in the US and SESA regions. An opposite although less pronounced effect is shown for the CB region. Strong La Niña events such as in 2010-2012 coincided with strong negative yield anomalies in the US and SESA regions in addition to unprecedent increases in global soybean price (van Garderen and

Mindlin, 2022; Hoerling et al., 2014). Future change in the frequency of ENSO phases is highly uncertain, and recent work highlighted that current climate models may be not well representing the associated physical processes (Seager et al., 2019, 2022; Wills et al., 2022). This has important consequences on projected future risk of spatially compound soybean production failures. The storyline approach can be an alternative framework to explore the potential impacts of such a spatially compound event in a warmer world (Goulart et al., 2022; Shepherd, 2019). This is particularly useful for future planning given the current

large physical uncertainties in climate models with respect to ENSO linked processes.

## 5 Conclusion

We presented a data-driven causal framework that links ENSO variability to extratropical SST patterns and local weather and soybean yield conditions in the US, Brazil and Argentina. We found that soybean yields are strongly linked to summer soil moisture in all three regions. Soil moisture during the summer is affected by antecedent soil moisture (spring) and extratropical

SST patterns, both of which are influenced by ENSO. Using causal networks to model such a system represents a relatively simple improvement to more typical regression analysis, enabling explicit hypothesis testing and thus a better understanding of cause-effect relationships. Our analyses indicate that a persistent la Niña's favor significantly lower yields in the US and SESA regions and only give a minor positive yield effect in the CB region. A persistent la Niña event of moderate magnitude (ENSO index of -1.5 SD) correspond to a reduction of -0.4 SD in soybean yields in both the US and SESA, substantially

constraining global soybean production.





## Appendix A: Additional figures

**(a)** Bootstrapped path coefficients for local climate effects

**(b)** Bootstrapped path coefficients for remote climate effects

**Figure A 1. Panel (a) shows bootstrapped path coefficient estimates based on 1000 draws for local climate effect models. Panel (b) shows bootstrapped path coefficients estimates based on 1000 draws for remote climate effect models.**







**Figure A 2. Correlation maps at both lagged and same season timesteps for Z500 variable against summer soil moisture. Stippling indicates statistical significance ($P_{FDR} < 0.1$) after having corrected for false discovery rate using the Benjamini/Hochberg correction.**



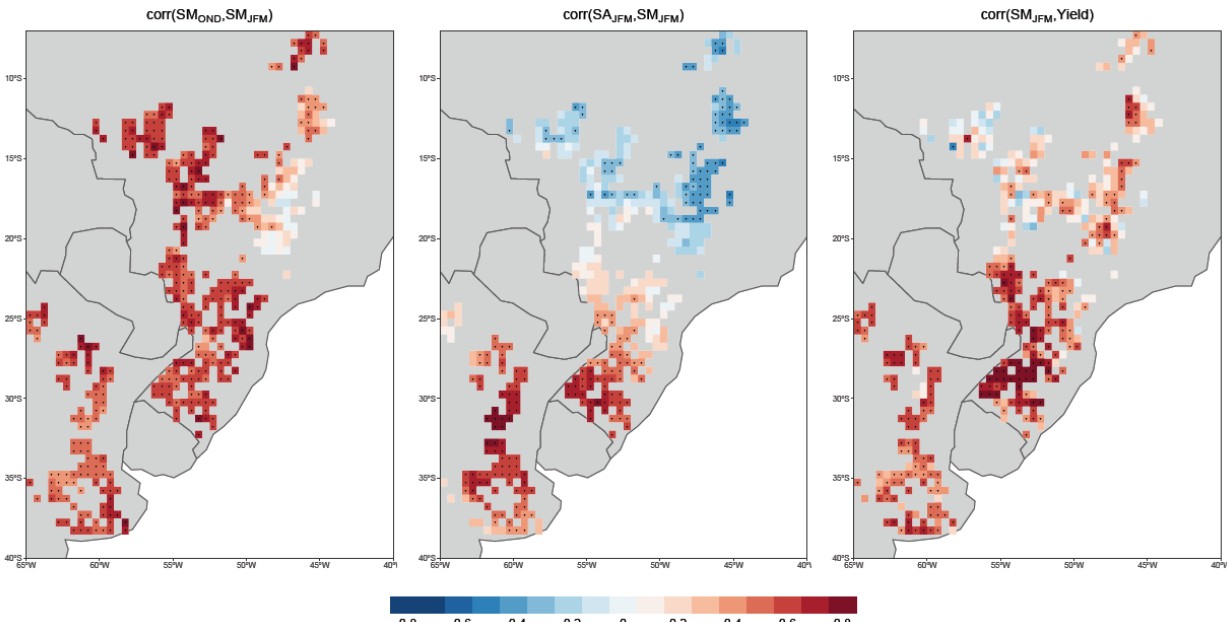

**Figure A 3. Grid-based correlation maps for crop yields, the SA pattern and antecedent (austral spring-OND) soil moisture against summer soil moisture (austral summer-JFM)**

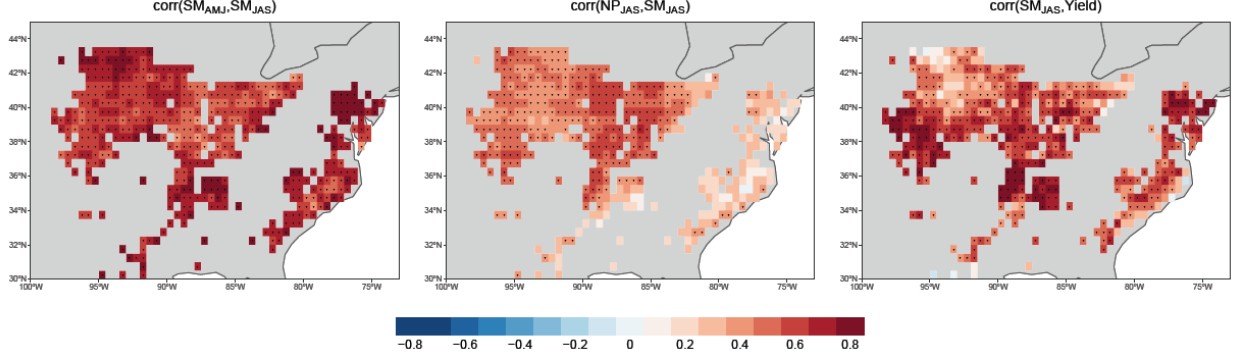

**Figure A 4. Grid-based correlation maps for crop yields, the NP pattern and antecedent (boreal spring-AMJ) soil moisture against summer soil moisture (boreal summer-JAS)**

*Code Availability.* The code is available from the corresponding author upon reasonable request.

*Data Availability.* Data used in this study are freely available in the cited literature.

*Author contributions.* RH, SV and DC designed the study. RH performed the analysis and wrote the initial draft of the paper. All authors contributed to the development of the analysis, the interpretation of the results and to writing the paper.

*Competing interests.* The contact author has declared that neither they nor their co-authors have any competing interests.





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
