# Peer review of "Persistent La Niña's drive joint soybean harvest failures in North and South America"

_EGUsphere, 2022_

## Author Response (AR2)

We would like to thank the reviewers and editor for their constructive comments, which, we believe, helped improving the quality of the submitted manuscript. Please find below a list of all relevant changes made in the manuscript followed by a point by point response to the reviewers and editor (responses in italic and bold). Referenced Line numbers refer to the tracked changes version of the manuscript.

**List of all relevant changes:**

1- We specified in the main text that the 3-month periods we consider correspond specifically to the typical soybean sensitive growth period in North and South America (L37).
2- We added to the main text an estimate of the total share of global production covered in our study (L87-88).
3- We added information on the land-atmosphere coupling and link between (moisture → heat) earlier in the manuscript (L155 -162).
4- We added additional information in the main text with respect to the implications of ENSO evolution (i.e. persistent vs developing La Niña conditions) on the co-occurrence of hot and dry conditions and joint crop failures in SESA and US regions (L349-358)
5- We added to the appendix composite maps of summer soil moisture, extreme heat and soybean yield anomalies for persistent La Niña vs developing La Niña years (Fig.A5,A6,A7)
6- We highlighted the potential limitations associated to our use of a static harvested area map and crop calendar when it comes to explained yield variability in central Brazil (393-394).
7- We added conditional independence claims implied by our causal diagrams to the supplementary material.
* * *
**Point-by-point response to the reviewers:**

**Referee 1:**

The authors build a framework that links direct and indirect effects of ENSO on global soybean production with a focus on North and South America. The authors use linear structural causal models (SEM) to quantify the impacts of spring and summer soil moisture as well as extreme heat on soybean yield anomalies in the main growing regions. In addition, ENSO variability is linked to extratropical SST patterns, local weather and soybean yields in Brazil, Argentina and the US.

General comments:

The paper is very well written and clearly structured. Findings are compared with results of similar studies and put into context. Using SEM models to link ENSO, SST and soil moisture, as well as soil moisture, heat and yields is an innovative approach that builds on previous research which only focused on parts of the causation chain. Limitations and potential future research are well described.

***We thank the reviewer for the overall positive evaluation of our study. In what follows, we provide a point-by-point reply to the reviewer constructive comments.***

The title, however, is misleading as the majority of the analysis refers to correlations between ENSO, SST, soil moisture and soybean yields in each of the three breadbaskets. Spatial correlations between the three production areas via ENSO conditions are only mentioned in the Discussion and not explicitly analyzed. Thus, I suggest a change of title or additional analysis of spatial correlations, e.g. of soil moisture conditions or crop yields between the regions.

*We agree with the reviewer that co-variability among the regions can be more explicitly addressed in the paper. Our results show that ENSO (OND) influences the SA pattern, summer soil moisture and soybean yields in South America whereas ENSO (AMJ) influences the NP pattern, summer soil moisture and soybean yields in the US. It follows that the co-occurrence of low soil moisture conditions and crop yields, for a given year, in both the US and SESA is favored when both ENSO (OND) and ENSO (AMJ) are in a La Niña state, hence the proposed title. Correlations between soil moisture conditions or crop yields between regions can fail to capture this dependence as the relationship across regions vary among years based on the background evolution of ENSO.*

*In what follows we provide some context. ENSO typically develops during boreal summer, peaks in boreal winter and decays in the following boreal spring. Developing La Niña years since 1950 have all been preceded by boreal winter El Niño conditions (Jong et al., 2020). It follows that a developing La Niña tend to favor opposite moisture and yield conditions over the SESA and US regions (Anderson et al., 2019).*

*To illustrate this changing dependence, we plot composites of summer soil moisture, extreme heat and crop yields for different ENSO evolutions. Persistent La Niña years are defined as years when both (OND) and (AMJ) ENSO indices are below -0.5. Developing La Niña years are defined following Jong et al. (2020) to remain consistent with the body of literature on the topic and to reflect one of the reviewer 2 comments.*

[Figure]

*Figure R1: Composites of Summer soil moisture, summer extreme heat and soybean yield anomalies for persistent La Niña years (indicated in the subtitle). Summer periods are JFM in the southern hemisphere and JAS in the northern hemisphere.*

[Figure]

*Figure R2: Similar to Fig. R1 but the subset additionally selects for years where both the southern hemisphere summer SA pattern & north hemisphere summer NP pattern are negative.*

[Figure]

*Figure R3: Similar to Fig. R1 but considering developing La Niña years as per (Jong et al., 2020).*

**Our composites show that persistent La Niña years favor consistently hot and dry summers in addition to low soybean yield anomalies in SESA and the US as suggested in the title (Figure R1). This is even more pronounced during persistent La Niña years accompanied by negative SA and NP patterns in respective austral and boreal summers (Figure R2). Developing La Niña years however show an opposite signal with wet and cool summers in SESA and hot and dry summers in the US (Figure R3). This emphasizes the role of ENSO evolution in regulating the dependence across regions. In our opinion, the title is still justified as it signals the central outcome of our analysis. If the reviewer is not convinced, we are willing to change our title to "Disentangling remote and regional drivers of joint soybean harvest failures in the Americas". Information with respect to the changing dependence between regions depending on ENSO evolution will be added to the text. In addition, we will make the composites available in the appendix material.**

Specific comments:

The authors state that crops are particularly vulnerable during the reproductive time and identify relevant months for South America and months for North America. I suggest specifying that this refers to soybeans as soil moisture sensitive growing periods differ between crops and regions. There are also differing definitions of sensitive growing periods. In addition to the papers that the authors cite, USDA has published crop calendars with slightly differing moisture sensitive growing periods: https://ipad.fas.usda.gov/ogamaps/cropmapsandcalendars.aspx

***We thank the reviewer for referencing the monthly USDA crop calendar. We will specify in the text that the 3-month periods we consider correspond specifically to the typical soybean sensitive growth period which includes flowering and grain filling stages. The monthly periods highlighted in the USDA crop calendar specifically mentions August for soybean in the US, January for soybean in SESA and finally December for soybean grown in CB. We compare models of standardized yield anomaly using our 3-month summer timeseries of heat and drought vs USDA specific month timeseries.***

[Figure]

*Figure R4: Model fitted Standardized yield anomalies based on 3-month aggregates of summer soil moisture and extremes heat vs 1-month aggregates based on the USDA calendar for the same variables.*

***Models show qualitatively and to a large extent quantitatively similar results except for the CB region where 3-month summer captures better yield variability compared to December soil moisture and extreme heat (Figure R4). Given this, we prefer to keep the summer definition of the sensitive period and highlight in the manuscript that such periods can be further detailed to better tackle specific crop physiological stages.***

Regarding the selection of counties, using rainfed production and harvested areas makes sense. It would be useful to provide an estimate of the total share of global production that the study covers in the end. The authors state that the US, Argentina

and Brazil account for 80% of global soybean production. After excluding a few relevant US states, is the share of global production in your study still significant?

*Soybean global supply vary among seasons but 80% is a good estimate when it comes to the US, Argentina and Brazil share of global soybeans supply over the last 20 years: https://resourcetrade.earth/?year=2020&exporter=842&category=87&units=value&autozoom=1. Our study assumes a fixed harvested area based on the year 2000. This is a limitation imposed by the MIRCA2000 dataset (Portmann et al., 2010) we used in our study to separate between irrigated and rainfed regions. The actual harvested area changes over time which makes an accurate estimate of the global production share in our selected regions difficult. However, taking the MIRCA2000 dataset as reference, we can provide the following estimates:*

- *The filtered rainfed area represents 95% of the total soybean MIRCA2000 harvested area in the Americas*
- *By further selecting for counties with at least 30 data points, we reduce the area to 72% of the total MIRCA2000 soybean harvested area*
- *By further omitting the few relevant US states, this number is reduced to 57%*

*This means that our study region covers around 46% of the global soybean production area given that the MIRCA2000 dataset is still representative of actual harvested areas. In our opinion, this is still a significant share of the global soybean supply. We will add this information to the manuscript.*

I have trouble to understand why soil moisture drives (assuming causality) extreme heat. I would assume the driver of soil moisture is heat. Please provide more information on the underlying land-atmosphere coupling you are referring to in the methodology paragraph (It is mentioned in the discussion later. I suggest referring to it already earlier in the manuscript).

*We thank the reviewer for the comment and will add information on the land-atmosphere coupling in the method section. In what follows we briefly touch upon the reason for setting up the causality in summer from soil moisture to extreme heat. Soil moisture and temperature are tightly coupled in the climate system. Higher temperatures will lead to more evapotranspiration and therefore reduce soil moisture. On the other hand, low soil moisture will reduce evapotranspiration which limits energy partitioning into latent heat, therefore increasing sensible heat and, in consequence, temperature. Although both these mechanisms are part of a feedback system, the first causal chain (heat -> Soil Moisture) dominates in an energy limited regime whereas the second (Soil moisture -> extreme heat) becomes increasingly important once soils are dry (Seneviratne et al., 2010). Summers in our study regions are characterized by a moisture limited regime (Lesk et al., 2021) hence the assumed directionality in our causal diagrams. This remains a qualitative choice which we will clarify in the main text. To study quantitatively causality between soil moisture and heat, one would potentially need a different set-up which considers high temporal resolution lead-lag relationships between moisture and heat.*

There is an important difference between correlation and causation, as I am sure the authors are well aware of. I also assume that the SEM methodology considers this. It would be helpful if the authors elaborated on this further, especially regarding the example soil moisture -> heat or heat-> soil moisture.

*The SEM methodology allows to test whether the independence claims implied by our causal diagram are consistent with observed data. For example, the diagram hypothesized for the local models (i.e. Fig. 2 in the submitted manuscript) implies the following independence claims:*

- *$SM_{spring} \perp Heat_{summer} \mid \{SM_{summer}\}$*
- *$SM_{spring} \perp yield_{anomaly} \mid \{SM_{summer}, Heat_{summer}\}$*
- *$RF_{summer} \perp Heat_{summer} \mid \{SM_{summer}\}$*
- *$RF_{summer} \perp yield_{anomaly} \mid \{SM_{summer}, Heat_{summer}\}$*

*Taking the first claim as example, it reads as: "$SM_{spring}$" and "$Heat_{summer}$" are independent ("$\perp$") given ("$\mid$") the conditioning set W = $\{SM_{summer}\}$. Translated into a statistical test, this implies that the effect of $SM_{spring}$ on $Heat_{summer}$ is not significantly different from zero once we control for the influence of $SM_{summer}$ on $Heat_{summer}$. SEM tests the full set of conditional independence claims and concludes whether a proposed causal hypothesis is consistent with the observed data. The conditional independence claims implied by our causal diagrams will be added to the supplementary material so that what is being tested is made more explicit.*

*With respect to (moisture -> heat or heat-> soil moisture), the quantitative notion of causality in our study is limited to the set of conditional independence claims we test. Given that we don't test explicitly for soil moisture -> Heat vs Heat -> soil moisture, the direction of this relationship remains qualitatively set.*

*The two alternative causal hypotheses (i.e., soil moisture -> Heat vs Heat -> soil moisture) are both consistent with the observed data. The set of conditional independence claims we test in the case of Heat -> Soil moisture is:*

- *$SM_{spring} \perp yield_{anomaly} \mid \{SM_{summer}, Heat_{summer}\}$*
- *$RF_{summer} \perp yield_{anomaly} \mid \{SM_{summer}, Heat_{summer}\}$*

*Model parameter estimation remains qualitatively similar among both causal hypotheses (Figure R5). Our preference is to keep the direction (Soil moisture -> Heat) to reflect important land-atmosphere feedbacks that are particularly impactful during*

***dry summers.***

[Figure]

*Figure R5: Model parameter estimation based on authors causal diagram (upper panel) and reviewer proposed causal diagram (lower panel).*

**References:**

Anderson, W. B., Seager, R., Baethgen, W., Cane, M. and You, L.: Synchronous crop failures and climate-forced production variability, Sci. Adv., 5(7), 1–10, doi:10.1126/sciadv.aaw1976, 2019.

Jong, B. T., Ting, M., Seager, R. and Anderson, W. B.: ENSO Teleconnections and Impacts on U.S. Summertime Temperature during a Multiyear la Niña Life Cycle, J. Clim., 33(14), 6009–6024, doi:10.1175/JCLI-D-19-0701.1, 2020.

Lesk, C., Coffel, E., Winter, J., Ray, D., Zscheischler, J., Seneviratne, S. I. and Horton, R.: Stronger temperature–moisture couplings exacerbate the impact of climate warming on global crop yields, Nat. Food, 2(9), 683–691, doi:10.1038/s43016-021-00341-6, 2021.

Portmann, F. T., Siebert, S. and Döll, P.: MIRCA2000-Global monthly irrigated and rainfed crop areas around the year 2000: A new high-resolution data set for agricultural and hydrological modeling, Global Biogeochem. Cycles, 24(1), n/a-n/a, doi:10.1029/2008gb003435, 2010.

Seneviratne, S. I., Corti, T., Davin, E. L., Hirschi, M., Jaeger, E. B., Lehner, I., Orlowsky, B. and Teuling, A. J.: Investigating soil moisture-climate interactions in a changing climate: A review, Earth-Science Rev., 99(3–4), 125–161, doi:10.1016/j.earscirev.2010.02.004, 2010.

**Referee 2**

Hamed and co-authors present an analysis framework to link crop yield anomalies to crop growing conditions and subsequently the underlying climate drivers in a causal chain of analysis. They build on past work to demonstrate that their method is relevant in the case of multiple crop yield shocks to soybeans in North and South America. The authors present a well written and well-motivated study with easy to interpret graphs. I thank the authors for the time and care that has gone in to the manuscript. I generally think the manuscript sound and I have only three minor comments and suggestions for the authors to consider.

*We thank the reviewer for their kind words and positive evaluation of our study. In what follows, we provide a point-by-point reply to the reviewer constructive comments.*

 Specific comments:

1. For central Brazil, the relative soybean growing seasons have changed over time with the increase in Safrinha cycle cropping. If you are using a static harvested area map and crop calendar to weight the climate anomalies and produce a regionally aggregated weather time series to relate to the regional crop yield time series, the change in dominance from traditional crop cycles to a safrinha soy-maize crop cycle may introduce error. Your approach is a reasonable enough as it is, but this limitation may be worth mentioning in the context of the smaller variance explained by climate variables in central Brazil as compared to SESA. South Brazil does not produce much Safrinha cycle soy-maize crop rotations, so the analysis in South Brazil would not be strongly affected by this.

   *Thank you for highlighting this. We will add this information to the revised manuscript and mention that future studies can further explore such aspects when studying soybean yield and climate variability in Brazil.*

2. The soybean growing season in the US (May-Oct) intersects typical ENSO development (~Jul) and decay (~Mar) such that one could develop reasonable hypotheses that the intersection of the soybean season with either a developing ENSO event (Jul-Oct) or the lagged effect of a decaying ENSO event (Apr - Jun) might affect the soybean growing season. Can your causal framework distinguish between these two different cases, and if so what do the conclusions say about whether we should be considering developing ENSO events, decaying ENSO events, or both when evaluating the effect of ENSO on summer-grown crops in the US? It would be helpful to clarify this, especially because the past literature you cite (e.g. Anderson et al. 2017a, 2017b, 2018) outlines the effects of ENSO primarily as developing events, although Jong et al. (2020) highlight the importance of antecedent SST anomalies in the west pacific for US summertime heat during La Niñas (https://journals.ametsoc.org/view/journals/clim/33/14/jcliD190701.xml).

*Our framework highlights the north pacific pattern as the direct driver of summer weather conditions in the US and therefore yield impacts in the region. To illustrate this relationship, we plot timeseries of the NP pattern, soybean yields and summer soil moisture in the US (Figure R6). In black, we mark persistent La Niña years (Niña events that persist into AMJ). In gray, we mark developing La Niña events following Jong et al. (2020).*

[Figure]

*Figure R6: NP pattern, summer soil moisture and yield anomalies in the US. Black dots highlight persistent La Niña events. Gray dots highlight developing La Niña events.*

*Negative NP pattern conditions occur during both developing and decaying (persisting) La Niña events which suggests that both should be considered when evaluating the effects of ENSO on summer grown crops in the US.*

*Composite maps based on developing and persistent ENSO events for summer soil moisture, extreme heat (EDD) and soybean yields both report hot and dry conditions in addition to low soybean yields over large US producing regions.*

[Figure]

*Figure R6: Composites of Summer soil moisture, summer extreme heat and soybean yield anomalies for persistent La Niña years (indicated in the subtitle). Summer periods are JFM in the southern hemisphere and JAS in the northern hemisphere.*

[Figure]

*Figure R7: Similar to Fig. R1 but considering developing La Niña years as per (Jong et al., 2020).*

***Nevertheless, we note that for persistent La Niña events, hot-dry and negative yield anomalies are more concentrated in the Midwest (A key soybean producing region) whereas the most eastern states show cool, wet and positive yield anomalies (Figure R1). In the case of developing La Niña events, the hot-dry and low yield anomalies are present over practically the entire US soybean producing region (Figure R3). We will add text in the manuscript to highlight the potential differences between developing and persisting La Niña events.***

Clarify what is meant by "persistent" La Niñas. Do you mean multi-year, or La Niña events that persist into AMJ?

***We mean La Niña events that persist into AMJ. We will clarify this in the manuscript.***

**Editor:**

This is a very interesting study with relevant findings.

***Dear Olivia,***

***Thank you for finding our study interesting and relevant. In what follows, we elaborate on your constructive comments.***

I have two questions regarding the relation between the yield and the soil moisture. How does your methodological approach deal with threshold behaviours of the link between soil moisture and yield and pot. changes in the sign of the link depending on the magnitude of the anomaly? How does your approach deal with an asymmetrical link between soil moisture and yield anomalies for + and - soil moisture anomalies?

*The method we apply (i.e. Structural equation modeling) assumes linear functional relationships between any two variables but this can be in principle extended to model non-parametric relationships (Bongers et al., 2021). To explore potential threshold behaviors and asymmetrical links between summer soil moisture and yield anomalies, we visualize scatter plots of the two spatially averaged variables for each region separately. Additionally, we add scatter plots of the two variables based on the full grid-cell based dataset. Using the full dataset allows to leverage a larger sample size that can potentially reveal details that might be masked in the spatially averaged time-series.*

[Figure]

*Figure R7: Scatter plots between yield anomalies and summer soil moisture for spatially averaged data over the different study regions separately. Linear (in red) and quadratic (in blue) fits over the data to reveal relationship tendencies in the data.*

[Figure]

*Figure R8: Scatter plots between yield anomalies and summer soil moisture for pooled data over the different study regions separately. Linear (in red) and quadratic (in blue) fits over the data to reveal relationship tendencies in the data.*

*Results show an overall increase in yield anomalies for higher levels of soil moisture for all regions consistent with the assumption we make in this study. This dominant linear relationship can be potentially explained by our focus on summer season soil*

*moisture which often reports lower absolute values compared to, for e.g. spring seasons. Still, for large values of summer soil moisture, we see the possibility of negative yield anomaly occurrence which is consistent with reported negative effects of excess moisture on crops. Given the low occurrence of such events in addition to our focus here on the effects of hot and dry conditions, we believe that assuming a linear functional form is sufficient and doesn't compromise much the estimation of the relationship between summer soil moisture and yield. We do note however in the text that soybean yields can also be impacted by excessive rainfall and soil moisture (L337), an aspect that can be further explored in future studies.*

I recommend one additional round of proofreading, there are some small grammatical errors and typos and formulations that could still be improved, e.g., line 3 soybean yield, or l29 increasingly growing --> growing

*Thank you, we will do this when submitting the revised version of our manuscript.*

*References:*

Bongers, S., Forré, P., Peters, J. and Mooij, J. M.: Foundations of structural causal models with cycles and latent variables, Ann. Stat., 49(5), 2885–2915, doi:10.1214/21-AOS2064, 2021.